# Detailed Molecular Interactions of Favipiravir with SARS-CoV-2, SARS-CoV, MERS-CoV, and Influenza Virus Polymerases In Silico

**DOI:** 10.3390/microorganisms8101610

**Published:** 2020-10-20

**Authors:** Mitsuru Sada, Takeshi Saraya, Haruyuki Ishii, Kaori Okayama, Yuriko Hayashi, Takeshi Tsugawa, Atsuyoshi Nishina, Koichi Murakami, Makoto Kuroda, Akihide Ryo, Hirokazu Kimura

**Affiliations:** 1Advanced Medical Science Research Center, Gunma Paz University, Shibukawa, Gunma 377-0008, Japan; rainbow_orchestra716@yahoo.co.jp; 2Department of Respiratory medicine, Kyorin University Hospital of medicine, Mitaka, Tokyo 181-8611, Japan; saraya@ks.kyorin-u.ac.jp (T.S.); h141@ks.kyorin-u.ac.jp (H.I.); 3School of Medical Technology, Faculty of Health Science, Gumma Paz University, Takasaki, Gunma 370-0006, Japan; okayama@paz.ac.jp (K.O.); hayashi@paz.ac.jp (Y.H.); 4Department of Pediatrics, Sapporo Medical University School of Medicine, Sapporo, Hokkaido 060-8543, Japan; tsugawat@sapmed.ac.jp; 5College of Science and Technology, Nihon University, Tokyo 101-0062, Japan; nishina.atsuyoshi@nihon-u.ac.jp; 6Infectious Disease Surveillance Center, National Institute of Infectious Diseases, Tokyo 162-8640, Japan; kmuraka@niid.go.jp; 7Pathogen Genomics Center, National Institute of Infectious Diseases, Tokyo 208-0011, Japan; makokuro@niid.go.jp; 8Department of Microbiology, Yokohama City University School of Medicine, Yokohama, Kanagawa 236-0004, Japan; aryo@yokohama-cu.ac.jp; 9Department of Health Science, Gunma Paz University Graduate School of Health Sciences, Takasaki, Gunma 370-0006, Japan

**Keywords:** favipiravir, COVID-19, SARS-CoV-2, influenza, RdRp, in silico

## Abstract

Favipiravir was initially developed as an antiviral drug against influenza and is currently used in clinical trials against severe acute respiratory syndrome coronavirus-2 (SARS-CoV-2) infection (COVID-19). This agent is presumably involved in RNA chain termination during influenza virus replication, although the molecular interactions underlying its potential impact on the coronaviruses including SARS-CoV-2, SARS-CoV, and Middle East respiratory syndrome coronavirus (MERS-CoV) remain unclear. We performed in silico studies to elucidate detailed molecular interactions between favipiravir and the SARS-CoV-2, SARS-CoV, MERS-CoV, and influenza virus RNA-dependent RNA polymerases (RdRp). As a result, no interactions between favipiravir ribofuranosyl-5′-triphosphate (F-RTP), the active form of favipiravir, and the active sites of RdRps (PB1 proteins) from influenza A (H1N1)pdm09 virus were found, yet the agent bound to the tunnel of the replication genome of PB1 protein leading to the inhibition of replicated RNA passage. In contrast, F-RTP bound to the active sites of coronavirus RdRp in the presence of the agent and RdRp. Further, the agent bound to the replicated RNA terminus in the presence of agent, magnesium ions, nucleotide triphosphate, and RdRp proteins. These results suggest that favipiravir exhibits distinct mechanisms of action against influenza virus and various coronaviruses.

## 1. Introduction

In December 2019, a new coronavirus disease (coronavirus disease 2019, COVID-19) emerged suddenly in China [1]. COVID-19 spread rapidly, resulting in a pandemic [2]. Over 34 million people were confirmed COVID-19-positive as of the end of September 2020 [2]. The agent is classified into genus *Betacoronavirus*, based on detailed genome analyses, and formally named severe acute respiratory syndrome coronavirus-2 (SARS-CoV-2) [3]. About 10–20% of COVID-19 cases may cause fever, fatigue, cough, and pneumonia, while the infection in some cases may result in inapparent symptoms [4]. However, some cases of COVID-19 may be complicated acute respiratory distress syndrome (ARDS) leading to death [5]. Thus, a need exists for the early development of new therapeutic drugs and applications of existing drugs for the treatment of COVID-19. To date, some antiviral agents including ciclesonide, remdesivir, and favipiravir have been tried to treat COVID-19 [6].

Inhibition of viral replication is assumed to be the mechanism for therapeutic agents for SARS-CoV-2. RNA-dependent RNA polymerase (RdRp) of SARS-CoV-2 is nonstructural protein 12 (NSP-12) and is similar to SARS-CoV RdRp [7]. NSP-7 and NSP-8 are known cofactors of SARS-CoV-2 RdRp proteins. Further, NSP-15 protein is an endonuclease from SARS-CoV-2 that plays an important role in the proofreading of viral RNA. Thus, these proteins may be targets for antiviral drugs for treating COVID-19.

Favipiravir (6-fluoro-3-hydroxypyrazine-2-carboxamide, Avigan^®^) was initially developed as an antiviral agent for the treatment of influenza. The mechanism of favipiravir is inhibition of viral RNA replication by inhibition of RdRp formed as a complex of PA protein, PB1 protein, and PB2 protein [8,9,10,11]. Subsequently, favipiravir was reported to have similar activity against RdRp proteins from RNA viruses other than influenza and showed efficacy for treatment of ebolavirus disease (EVD), Lassa fever, and norovirus infections [12,13,14]. Thus, favipiravir might also be effective against some RNA virus infections, and is currently in clinical evaluation as a treatment for COVID-19 [6]. However, no detailed molecular interactions between favipiravir and SARS-CoV-2 proteins, such as RdRp and NSP-15 endonuclease, are exactly known. With this background, we performed an in silico study regarding the molecular interactions among favipiravir, influenza RdRp, SARS-CoV-2 RdRp, SARS-CoV RdRp, Middle East respiratory syndrome coronavirus (MERS-CoV) RdRp, and the coronaviruses’ NSP-15 endonuclease.

## 2. Materials and Methods

### 2.1. Structural Modeling

Sequences of the RdRp proteins of influenza A/Northern Territory/60/1968/H3N2 (PDBID: 6qnw), bat influenza A polymerase (PDBID: 6szu), and SARS-CoV-2 (PDBID: 6 m71, 7bv2) were downloaded from Protein Data Bank Japan (https://pdbj.org/). We also downloaded NSP-15 of SARS-CoV-2 (PDBID: 6x1b), SARS-CoV (PDBID: 2h85), and MERS-CoV (PDBID: 5yvd) from Protein Data Bank Japan. Sequences of the RdRp proteins of influenza A/California/07/2009(H1N1) (Protein ID: YP_009118630.1, YP_009118628.1, YP_009118631.1), RdRp proteins of MERS-CoV (Protein ID:YP_009047223.1, YP_009047220.1, YP_009047219.1) were downloaded from NCBI (https://www.ncbi.nlm.nih.gov/protein/). The three-dimensional (3D) structure of influenza H3N2 PB1 protein was constructed based on data available on the structure of the B chain of influenza H3N2 RdRp protein.

The homology model of influenza A/California/07/2009(H1N1) RdRp protein was built with the template structure of bat influenza A polymerase (PDBID: 6szu) using MODELLER 9.23 software on the Windows operating system [15]. The query sequences from A/California/07/2009(H1N1) strain were searched to find related protein structures for use as templates for the BLAST (Basic Local Alignment Search Tool) program (https://blast.ncbi.nlm.nih.gov/Blast.cgi) against PDB [16]. The 3D model obtained was evaluated by Ramachandran’s map using RNA Annotation and Mapping of Promoters for the Analysis of Gene Expression (RAMPAGE) and by the Swiss Protein Databank Viewer (SPDBV) 4.10 software [17,18]. Nucleotide triphosphates (NTPs) ions were manually inserted to the RdRp proteins referring to bat influenza A polymerase (PDBID:6szu). We used coronavirus RdRp protein that contained NSP-12, NSP-7, and NSP-8. The model of SARS-CoV-2 RdRp protein with NTP and ions was manually constructed from SARS-CoV-2 RdRp (PDBID: 7bv2). The homology model of SARS-CoV RdRp protein and MERS-CoV RdRp protein was also built with the template structure of SARS-CoV-2 RdRp protein (PDBID: 7bv2). NTPs ions were manually inserted into SARS-CoV RdRp proteins and MERS-CoV RdRp proteins referring to SARS-CoV-2 RdRp proteins. The homology model for NSP-15 of SARS-CoV-2 was built as previously described [19]. NSP-15 of SARS-CoV-2 (PDBID: 6x1b) were downloaded from Protein Data Bank Japan.

The 3D structure of favipiravir (Accession Number: DB12466) used for the docking simulation analysis was obtained from the DRUGBANK database (https://www.drugbank.ca/), including the structural data. The putative structure of favipiravir ribofuranosyl-5′-triphosphate (F-RTP) was manually reconstructed using Molview v2.4 [20]. Favipiravir ribofuranosyl-5′-monophosphate (F-RMP) was manually reconstructed as previously described [10].

### 2.2. Docking Simulation

Computational simulation of the molecular recognition process was performed using AutoDock Vina 1.1.2 software according to software instructions [21]. Before docking compounds on the target, the protein was edited using AutoDockTools 1.5.6. Polar hydrogen atoms were added to amino acid residues, and Gasteiger charges were assigned to all atoms of the protein. The protein in PDBQT format was then used as an input to AutoDock Vina. The grid box for analysis was set to a size that included the entire protein region. Detailed procedures used for docking simulations have been previously reported [19,21,22]. Top 20 score poses were evaluated by using PyMOL 2.3.4 to visualize protein–ligand interactions. We evaluated binding energy, which was simply provided by programs. The active sites in influenza PB1 protein were added manually; these included Ser443, Asp444, and Asp445 based on findings in a previous report [23]. Similarly, the active site of SARS-CoV-2 RdRp protein was added manually and included residues Ser 759, Asp760, and Asp761 [24]. The active site of SARS-CoV RdRp and MERS-CoV RdRp was added manually including a similar motif in SARS-CoV-2 RdRp. Among the ligands bound to the active site, those associated with a root mean square deviation of 2 or more compared to values obtained prior to binding analysis were excluded from further consideration.

## 3. Results

### 3.1. Molecular Interactions between Favipiravir and F-RTP with Influenza PB1 Proteins

Neither favipiravir nor F-RTP bound to the active site of influenza virus subtype H1N1 or H3N2 PB1 proteins. Interestingly, F-RTP bound to sites in both the H1N1 and H3N2 PB1 proteins that were implicated in double-strand RNAs synthesis pathways. The chemical binding energy between F-RTP and the two PB1 proteins was estimated at −6.1 and −6.0 kcal/mol, respectively (Figure 1a,b). The F-RTP binding site on the H3N2 PB1 protein included Gln124, Arg249, Glu256, and Met411; the F-RTP binding site on the H1N1 PB1 protein included Asp743, Glu934, Arg940, Thr1009, Lys1010, and Lys1182.

### 3.2. Molecular Interactions between Favipiravir and F-RTP with SARS-CoV-2/SARS-CoV/MERS-CoV Replication-Associated Proteins RdRp and NSP-15 Endonuclease Alone

First, we examined molecular interactions using favipiravir/F-RTP and various proteins alone. As shown in Figure 1c, F-RTP bound directly to the active site of the SARS-CoV-2 RdRp; Asp760 was the critical amino acid residue facilitating this interaction. The binding energy between F-RTP and the SARS-CoV-2 RdRp was −6.6 kcal/mol. Uncharged favipiravir also bound to the SARS-CoV-2 RdRp at Asp760 and Asp761 with a binding energy of −4.0 kcal/mol. F-RTP bound to Ser679 in the active site of the SARS-CoV RdRp protein with a binding energy of −6.4 kcal/mol (Figure 1d). Similarly, F-RTP bound to the active site of MERS-CoV RdRp at Ser 678 and Asp680 with a binding energy of −7.3kcal/mol (Figure 1e). Both favipiravir and F-RTP have the potential to inhibit RNA polymerization catalyzed by SARS-CoV-2 RdRp, SARS-CoV RdRp, and MERS-CoV RdRp. In contrast, F-RMP did not interact with SARS-CoV-2 NSP-15, SARS-CoV NSP-15, and MERS-CoV NSP-15 (data not shown).

### 3.3. Molecular Interactions among Various Polymerases, Nucleotide Triphosphate (NTP), Magnesium Ion, and Favipiravir/F-RTP

We also analyzed molecular interactions using various viral polymerases, NTP, magnesium ions (Mg^2+^), and favipiravir/F-RTP. F-RTP bound to termini of replicated RNA of influenza RdRp protein with a binding energy of −9.8kcal/mol (Figure 2a). Similarly, F-RTP bound to termini of replicated RNA of all coronavirus RdRp proteins. Binding energy among the termini of the replicated RNA by SARS-CoV-2, SARS-CoV, and MERS-CoV were estimated as −8.4, −7.5, and −8.9 kcal/mol, respectively (Figure 2b–d).

## 4. Discussion

We showed in the present in silico study that F-RTP (the active form of favipiravir) could bind to RdRp active sites of SARS-CoV-2, SARS-CoV, and MERS-CoV in the presence of the agent and protein. Moreover, the F-RTP bound to the replicated RNA termini in the presence of the agent, magnesium ions, nucleotide triphosphate, and RdRp proteins. Conversely, F-RTP did not bind to PB1 (RdRp) active sites of influenza virus H1N1 in the presence of agent and protein. Further, F-RTP may bind to the tunnel of the PB1 protein leading to the inhibition of the replicated RNA passage. Thus, F-RTP displays distinct pharmacological effects on various coronaviruses and influenza virus subtype AH1N1.

Favipiravir is a nucleic acid analog and was developed as a therapeutic drug to be used to treat influenza [25]. The drug was approved in Japan in 2014 and was stockpiled for use in the event of a new influenza epidemic. Favipiravir is activated by intracellular conversion to F-RTP by hypoxanthine-guanine phosphoribosyltransferase [8]. Previous reports revealed that the actions of favipiravir/F-RTP against influenza involves the termination of genome replication, although the detailed molecular interactions between the drug and the PB1 polymerase had not been fully elucidated. Favipiravir is currently undergoing clinical evaluation for use in treating COVID-19 in some countries, including Japan. Some reports suggest that favipiravir is effective in this setting [26,27], although the molecular mechanisms underlying drug efficacy against SARS-CoV-2 have not been fully explored. To date, a few antiviral agents against coronaviruses have been approved. However, our results suggest favipiravir may show antiviral activity against SARS-CoV-2, SARS-CoV, and MERS-CoV, though the present study was purely in silico. Thus, further clinical studies may be needed to demonstrate efficacy.

We examined the presence of the agent and coronavirus RdRp proteins and also the presence of the agent, magnesium ions, nucleotide triphosphate, and the viral RdRp proteins. The agent could bind to RdRp proteins and could inhibit genome replication. A previous report showed that the agent inhibits SARS-CoV-2 in vitro [28]. This study suggests that inhibition of genome replication is termination [28]. However, detailed molecular interactions between the agent and the viral replication systems may not currently be known. Thus, the present molecular pharmacological results may be the first observations.

Further, previous reports suggest that the antiviral effect of the agent was premature termination of genome replication [8,29,30]. In our study, F-RTP could bind to the replicated RNA termini of influenza RdRp proteins, suggesting that the inhibition of the genome replication mechanism of influenza virus is termination. This may be compatible with earlier reports [8,29,30]. The present in silico study suggests that F-RTP does not bind to PB1 protein active sites of influenza virus subtype AH1N1 in the presence of agent and protein, but does suggest that F-RTP may bind to the tunnel of the protein. This binding may result in inhibition of replicated RNA passage. Based on the results and speculation, we suggest that favipiravir may exhibit two mechanisms of antiviral activity. This observation may also be a first.

The half-maximal effective concentration (EC_50_) of favipiravir against SARS-CoV-2 is 61.88 μΜ (9.4 μg/mL), which is comparable to Ebola virus (10.8–63 μg/mL) but higher than EC_50′_s of 0.030–0.46 μg/mL for influenza virus [25,31,32,33]. Thus, the clinical dose of favipiravir for COVID-19 will be higher than the dose for influenza but comparable to the dose for Ebola hemorrhagic fever [34]. However, the underlying cause of differences in EC_50_ for influenza and SARS-CoV-2 is unknown. The difference in mechanisms of action of favipiravir in influenza and SARS-CoV-2 may explain this difference in EC_50_.

Favipiravir binding to proteins other than SARS-CoV-2 RdRp has been investigated [35,36]. NSP-15 protein is an endonuclease from SARS-CoV-2 which plays an important role in the replication of viral RNA. We previously reported that ciclesonide inhibits viral replication in SARS-CoV-2 by binding to active sites of NSP-15 [19]. Hence, we also examined a docking simulation for interactions between favipiravir/F-RTP and NSP-15. However, favipiravir/F-RMP did not bind to these active sites. Thus, NSP-15 is not involved in differences in EC_50_ of favipiravir between influenza virus and SARS-CoV-2.

In conclusion, we found, in silico, that favipiravir/F-RTP could bind to active sites of coronavirus RdRp proteins and replicated RNA termini. We also showed that F-RTP binds near the tunnel of influenza RdRp protein. Distinct mechanisms underlying favipiravir-mediated interactions with influenza RdRp and coronavirus RdRp may help explain the need for different doses of the drug for effective clinical responses for treating influenza vs. SARS-CoV-2 infections.

## Figures and Tables

**Figure 1 microorganisms-08-01610-f001:**
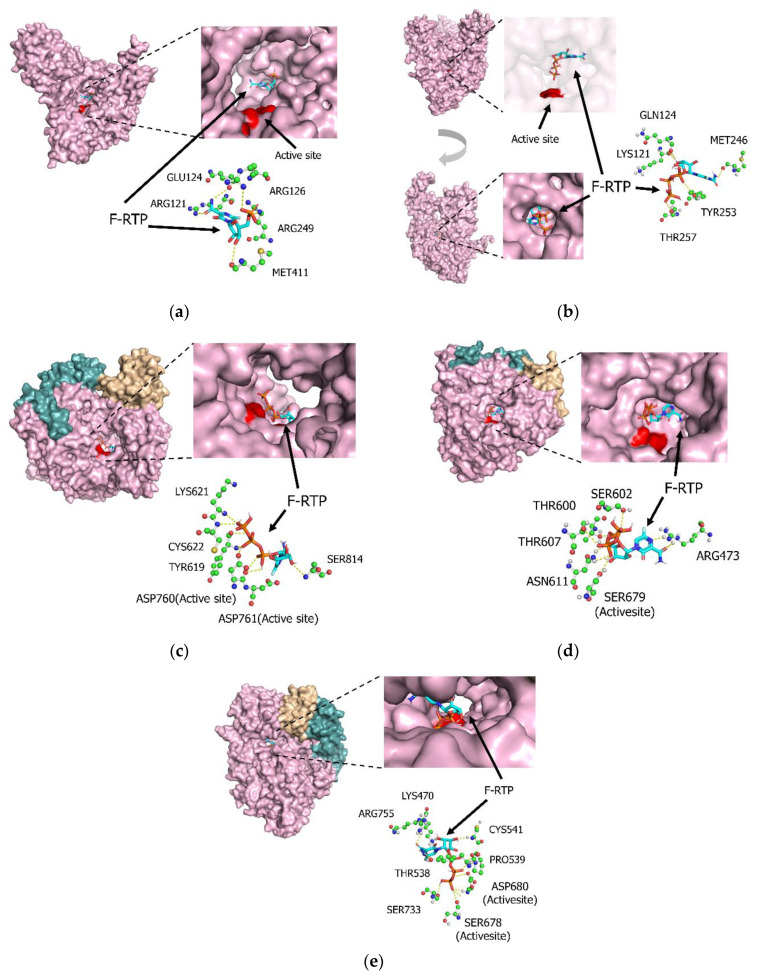
Detailed interactions between favipiravir ribofuranosyl-5′-triphosphate (F-RTP) and the active sites (red-colored regions) of (**a**) influenza H3N2 PB1 protein, (**b**) influenza H1N1 PB1 protein, (**c**) severe acute respiratory syndrome coronavirus-2 (SARS-CoV-2) RNA-dependent RNA polymerases (RdRp), (**d**) SARS-CoV RdRp, and (**e**) Middle East respiratory syndrome coronavirus (MERS-CoV) RdRp. Three-dimensional configurations of F-RTP and proteins were constructed with space-filling or stick models.

**Figure 2 microorganisms-08-01610-f002:**
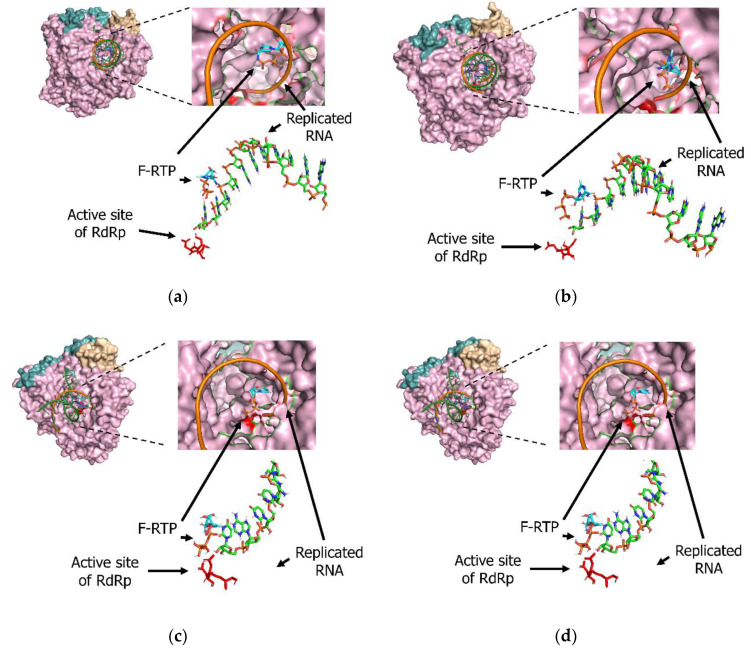
Detailed interactions among favipiravir ribofuranosyl-5′-triphosphate (F-RTP), nucleotide triphosphate (NTP), ions, and the active sites (red-colored regions) of (**a**) influenza H1N1, (**b**) severe acute respiratory syndrome coronavirus-2 (SARS-CoV-2), (**c**) SARS-CoV, and (**d**) Middle East respiratory syndrome coronavirus (MERS-CoV) RNA-dependent RNA polymerases (RdRp). Three-dimensional configurations of F-RTP, NTP, ions, and RdRp protein were constructed with space-filling or stick models.

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
