# Peer review of "Detailed Molecular Interactions of Favipiravir with SARS-CoV-2, SARS-CoV, MERS-CoV, and Influenza Virus Polymerases In Silico"

_microorganisms, 2020, doi:10.3390/microorganisms8101610_

Round 1

Reviewer 1 Report

This article by Mitsuru Sada et al. detailled in a very interesting maneer their in silico evaluation of the binding activity of Favipiravir on SARS-CoV2 and Influenza.

I think that this article is very interesting but has to be effectively completed in order to enhance its quality.

Major : 

As it was clearly described and could give information allowing interpratation of the obtained results, the authors have to developp the genetic comparison between respiratory virus especially between Coronaviridae, as SARS-Cov1 and MERS-CoV  share high similarity with SARS-CoV2.

Following the previous comment, it is necessary that the authors give results of a similar approach on other major coronaviruses, as MERS and SARS1. In absence of these detailled, the approach seem to not be complete.

Minor 

All used softwares have to be referenced, including the version and the developper.

Line 118 : interaction between NSP-15 and F-RTP has to be developped, as the authors concluded on it in the last part of the discussion.

Reviewer 2 Report

The manuscript by Sada et al describes a docking study of the nucleoside analog favipiravir triphosphate in resent cryo-EM structures of the influenza virus heterotrimeric RNA polymerase and the SARS-CoV-2 heterotrimeric RNA polymerase. The authors report that favipiravir triphosphate can be docked in the active site of the SARS-CoV-2 RNA polymerase, while it cannot be docked in the active site of the influenza virus RNA polymerase. The latter contradicts the experimental study performed by Goldhill et al 2018 PNAS, which the authors fail to discuss. The authors also fail to discuss the extensive favipiravir work by Shannon et al in the context of SARS-CoV-2. Finally, and more importantly, the authors i) overlook various basic molecular biology aspects of the enzymes that they are studying, ii) use a flawed computational approach and fail to report key details, and iii) only offer preliminary findings and no experimental data to support their conclusions. I have tried to provide constructive comments below, but unfortunately I do not see how this study would advance our understanding of favipiravir use in the current pandemic.

Major points

Why was PDB file 6qnw used for the influenza virus polymerase simulation experiments? 6qnw represents an inactive apo complex, which does not contain the viral promoter, a template or NTP in the active site. So, this complex gives the authors a suboptimal starting point for where the incoming NTP is bound.

Why did the authors only use the B chain, i.e. the PB1 subunit? The PB1 subnunit is insoluble by itself, can only function when bound to the other two subunits, and the previous favipiravir study cited by the authors (Goldhill et al PNAS) shows that favipiravir resistance involves a mutation in the PA subunit. The approach used by the authors thus appears to ignore the basic molecular biology of enzyme.

Why did the authors build a model of A/PR/8/34 H1N1? Why not use an actual, solved influenza polymerase structure? There are several (for flu A, B, C and D) out there that contain at least a template and often also an NTP and Mg2+ in the active site.

No information is given on which structure was used for the SARS-CoV-2 RNA polymerase or how the model was built.

What were the exact parameters for the docking approach? Did the authors optimize it for the two polymerases or just use the standard settings?

No experimental data (e.g. binding or NTP incorporation) is presented to support the in silico findings.

Line 98-100: What do the authors mean with “added”? And why were motif A and C added for the influenza virus polymerase but only motif C for the SARS-CoV-2 RNA polymerase?

Why do the authors assume that nsp15 can bind free F-RTP when its substrate is single-stranded RNA? This suggests a poor understanding of the molecular biology involved. They should have tested an ssRNA with an F-RMP in the RNA chain in their docking.

Resolution of the figure is very poor and it is impossible to see what the authors are indicating.

Line 105-107: It is unclear what “Neither favipiravir nor F-RTP bound to the active site of influenza virus subtype H1N1 or H3N2 PB1 proteins. Interestingly, F-RTP bound to sites in both the H1N1 and H3N2 PB1 proteins that were implicated in double-strand RNAs synthesis pathways” means. The active site of the RNA polymerase accommodates a template RNA and NTP. That the authors do not find F-RTP binding in this site is worrying – it should bind there as we know it is incorporated from other studies. It is also unclear where the dsRNA analysis comes from as it was not explained in the methods.

Line 108: How was the chemical binding energy between F-RTP and the two PB1 proteins estimated? Are these simply the values that the docking program provided? Did the authors do any controls?

Minor points

Line 61 and 69: the RdRp of influenza consists of three subunits, not just PB1. See e.g. te Velthuis et al 2016 Nat Rev Micro. This should be corrected.

The authors should introduce that the main polymerase activity of SARS-CoV-2/SARS-CoV is contained in nsp12, but the complex consists of at least two more subunits: nsp7 and nsp8.

Line 66: Favipiravir was demonstrated to inhibit SARS-CoV-2 infection and SARS-CoV RNA polymerase activity. See Shannon et al 2020: https://pubmed.ncbi.nlm.nih.gov/32511380/

Line 67: the authors should explain why it is important to consider the interaction of nsp15 with favipiravir? For the general reader it is unclear that nsp15 is important for the proofreading function of the replication complex and that it may specifically excise nucleoside analogs.

The text contains a lot of grammatical errors or poorly constructed sentences, including:

Line 47-48: “resulted a pandemic declared” is confusing

Line 48: “over 25 million individual were confirmed” is confusing

Line 60: “via inhibition viral RNA replication” is confusing

Reviewer 3 Report

In this manuscript, the authors conducted in silico studies to elucidate detailed molecular interactions between favipiravir and the SARS-CoV-2 and influenza RNA dependent RNA polymerases (RdRp). Their results suggest that both favipiravir and F-RTP have the potential to inhibit the activity of SARS-CoV-2 RdRp. To my impression, the manuscript is presented in a well-organized and logical manner. All the experimental results obtained from their studies show reasonable consistency. In addition, these studies provide insightful knowledge of Favipiravir and will contribute to explain why different doses of this drug are required for effective clinical responses. I would therefore strongly recommend this manuscript for publication in Microorganisms

Round 2

Reviewer 1 Report

The manuscript was well improved and now answer all the submitted questions.

The manuscript is now, for me, suitable for publication.

Author Response

 Thank you very much for reviewing this manuscript. We improved this manuscript to address the concerns of other reviewers.

Reviewer 2 Report

The manuscript is much improved by the authors and I commend them for their work. However, I still have one major issue and that is that the authors still used different RdRp-template complexes for their study and this affects their modelling and thus conclusions. Please see specific points below.

  1. Line 105 and 159-161: The 4wsb structure is in a pre-initiation conformation and does not contain a template or NTP in the active site. So this complex is very different from the SARS-CoV-2 complex 7bv2, which does contain a template and NTPs, and it is thus not correct to conclude that the binding mechanism is different (effectively, the authors are comparing apples and pears). Why not use the relevant complexes from https://pubmed.ncbi.nlm.nih.gov/32304664/ or use coordinates from the influenza B virus polymerase, which is pretty much identical to the influenza A virus polymerase and also contains a template and NTPs https://pubmed.ncbi.nlm.nih.gov/31160782/
  2. The BioRxiv reference for ref 28 may need to be updated. Please cite Shannon et al Nature Comm, https://pubmed.ncbi.nlm.nih.gov/32943628/
